# Forest Fire Risk Forecasting with the Aid of Case-Based Reasoning

Nikita Dorodnykh, Olga Nikolaychuk , Julia Pestova and Aleksandr Yurin *

Matrosov Institute for System Dynamics and Control Theory, Siberian Branch of Russian Academy of Sciences (ISDCT SB RAS), 664033 Irkutsk, Russia
* Correspondence: iskander@icc.ru

**Abstract:** Forest fire is one of the serious threats to the population and infrastructure of Irkutsk Oblast because its territory is heavily forested. This paper discusses the main stages of solving the problem of forecasting the risk of forest fires via a case-based approach, including data preprocessing, formation of a case model, and creation of a prototype of a case-based expert system. The main contributions of the paper are the following: a case model that provides a compact representation of information about weather conditions, vegetation type, and infrastructure of the region in relation to the possible risk of a wildfire; a case-base containing information about wildfires in Irkutsk Oblast for the period from 2017 to 2020; and a methodology for creating prototypes of case bases providing the transformation of decision tables of a special type. The approbation of the approach was carried out for separate forest districts, namely Bodaibinsk and Kazachinsk-Lena. The accuracy score was used for the evaluation of the results of forecasting the risk of wildfires. The average score value reached 0.51. The evaluation results revealed that application of the case-based approach can be considered as the initial stage for deeper investigations with the use of different methods (data mining, neural networks) for more accurate forecasting.

**Keywords:** hazard of forest fires; wildfire; forest quarters; forecasting; case-based reasoning; data analysis; Baikal natural territory; Irkutsk Oblast

## 1. Introduction

Wild and man-made fires remain a serious problem all over the world [1–6]. They have a detrimental effect on the composition and structure of fauna and flora and on the quality of air, soil, and water [7,8], which generally leads to degradation of ecosystems [9]. In addition, they are a serious threat to people and infrastructure [10,11]. Forest fires are caused by natural and climatic conditions (in particular, dry thunderstorms), careless handling of fire by the local population when picking berries and mushrooms, burning of grass, etc., while the main causes very often depend on the climatic or socio-cultural characteristics of the regions [12–14].

Effective monitoring and forecasting of their development is the basis for competent resource planning and informed decision making [15,16], including through the use of remote sensing methods [14,17]. This task is especially relevant for Irkutsk Oblast, the territory with the highest forest cover (78%) among the subjects of the Russian Federation, where fire-hazardous coniferous plantations predominate (more than 90% of the entire area covered by forest) [13].

To solve the forecasting task, different methods and tools based on machine learning [8,18], decision-making methods [14], fuzzy logic [19], clustering analysis [5], etc., are used. All of them require certain conditions and data preprocessing.

In this paper, we apply a case-based approach [20] from our experience for solving forecasting tasks in the fields of petrochemistry and technogenic emergencies [21,22]. The use of case-based reasoning (CBR) is considered in the context of solving the task of forecasting the risk of wildfires for Irkutsk Oblast, including data preprocessing, formation

of a case model, creation of a prototype of a case-based expert system and its debugging, integration of a prototype into a forest fire monitoring web service, and its evaluation.

CBR provides the effective use of accumulated experience in the absence of in-depth knowledge, using only superficial knowledge, i.e., knowledge about the visible relationships between events and facts in the domain. In this case, there is a direct mapping of the domain model, presented, for example, in a conceptual or set-theoretic form, into a case model, excluding complex training procedures, analysis, and evaluation of the parameters required for the employment of machine learning methods.

The main contributions of this paper that determine its novelty are the following:

- A domain model and a case model that provide a compact representation of information about weather conditions, vegetation type, and infrastructure of the region in relation to the possible risk of a wildfire. Information from various sources was used to build these models: GIS data, satellite images, weather stations data, literature, statistics, forest regulations, and expert assessments;
- A case base containing information about 2240 fires in Irkutsk Oblast for the period from 2017 to 2020. This case base was formed on the basis of the database that includes more than 45,000 records describing information about thermal points identified as a result of analysis of satellite images;
- A methodology for creating prototypes of case bases, providing the transformation of decision tables of a special type. The methodology includes the use of an original tool, namely personal knowledge base designer (PKBD) [23], and contains the following steps: preparing CSV files with decision tables for PKBD import, importing CSV files containing information for cases with the creation of a knowledge base, and testing imported data by queries with subsequent correction of the data. This methodology is a specialization of the approach from [24] for case bases;
- The results of evaluating the effectiveness of solving the task of forecasting the risk of wildfires based on CBR. The accuracy score was 0.51, which is lower than that of similar studies, but it can be further improved, including by clarifying the weights of the properties used in the case-retrieval procedure.

For Irkutsk Oblast, the study of forest fire forecasting based on CBR was conducted for the first time. These results were obtained at the first stage of our study. In the future, we plan to apply other techniques to improve the quality of forecasting.

The paper is organized as follows: Section 2 briefly describes the background, while Section 3 presents our results. Section 4 contains a discussion and concluding remarks.

## 2. Background

### 2.1. Forecasting the Risk of Forest Fires

An analysis of recent works in the field of fire forecasting [8,14,18,19] has shown a fairly large number of different methods and techniques for forest fire susceptibility and risk mapping. To solve this problem, probabilistic-based models such as Maxent models, fuel moisture content (FMC), and fire area simulator (FARSITE) [25–27] are proposed. The following methods are also used: machine learning [8,18,28] (including artificial neural networks (ANNs) [29–33]), fuzzy logic [14,19,34–36], clustering analysis [5], GIS-based multi-criteria decision analysis (MCDA) techniques, analytical hierarchy process (AHP) [15,19,37–39], fuzzy AHP [40], frequency ratio [41], logistic regression [19,29,31,42,43], Bayesian modeling [3], and CBR [44,45].

At the same time, CBR [20] allows one to combine parameters with quantitative (numerical) and qualitative (verbal) values. The main advantage of its application is the possibility of obtaining results without the model training stage. CBR is usually used to obtain preliminary results in order to further justify the use of other methods that require deeper research of the domain, but it also provides more accurate assessments [45].

The result of applying these methods and models is a risk assessment, for which various scales are used, in particular, the Canadian System of Weather Indices for Forest Fires (CFFWIS) [46]. However, the direct application of the described methods and scales

requires consideration of the individual properties of initial data and the studied areas [47]. These properties have the constant and dynamic character and influence the ability of ignition, spread rate, control difficulty, and fire impact [8]. The main requirements for wildfire occurrence and spread [48] are favorable climatic circumstances, the presence of fuel, its spatial continuity, and a source of ignition. Weather factors, human behavior, vegetation traits, fuel availability, and topography are among the most essential [49]. Some sources also propose division of the factors into groups [8]: (i) topography (or multi-class landscape indicators [6]), (ii) climatic, and (iii) fuel. Based on these works, it is possible to form a general list of factors that can be used to predict the development of a wildfire: height above sea level; terrain slope; topographic humidity index; distance from urban areas; average annual temperature; land use; distance from roads; average annual precipitation; distance to the river; air temperature (average daily and maximum); dates of average daily temperatures passing through threshold limits; dates of onset and descent of stable snow cover; relative humidity (average daily and minimum); lack of air humidity; the number of days with a relative humidity of $\leq 30\%$ in one of the observation periods for a certain period; the annual precipitation period; the number of days with rain; the dryness index; the weather regime; the number of days with thunderstorms, etc.

In the context of this research, we can single out studies focused on taking into account the peculiarities of the climatic zones of the Russian Federation and Irkutsk Oblast. We aim to provide an improvement of the scale for assessing fire hazard classes of forests depending on weather conditions [50] in order to introduce new factors, in particular, humidity indicators [51]. We also take into account regional characteristics [52,53] as well as existing methods for assessing fire safety [54]. The works considered above were used to analyze the domain and identify factors affecting the assessment of wildfire hazard.

### 2.2. Related Works: Using CBR for Forest Fires Forecasting

The review of the related research revealed two papers most relevant to the present study [44,45]. The first one [44] proposes a method to overcome the limitations of existing forest fire risk rating assessment methods based on remote sensing data and case-based reasoning principles. The method uses a combination of some dynamic and static indexes to characterize the potential fire environment. The authors built the fire risk rating spatial distribution maps for the DaXingAn Mountains of China using the lightning-caused forest fire risk rating assessment method during 2000–2006. The quantitative estimate of the effectiveness of the method by comparing actual and forecast data presented in the work showed 68.8% effectiveness of the forecast for a dataset with 560 cases.

The second paper [45] focuses on the development of software, namely the WeVoS-CBR system (ver. 1, Emilio Corchado, University of Salamanca, Salamanca, Spain), which is used to predict the evolution of forest fires. The case-based reasoning methodology combined with a summarization of the SOM ensembles algorithm was implemented. The WeVoS-CBR system successfully predicted the evolution of forest fires in terms of the probability of finding fires in a certain area. However, in [45], the use of CBR allowed one to achieve 58% good predictions for a dataset with 2000 cases and 70% for a dataset with 5000 cases.

Below, we use these works to compare and evaluate our results.

### 2.3. Case-Based Approach: Basic Principles

Case-based reasoning (CBR) [20,55] is a decision-making methodology that allows one to reuse and adapt (if necessary) previously obtained solutions to similar problems based on the principle of decision making "by analogy". The basic concept views a case as a structured representation of accumulated experience in the form of data and knowledge that provides its subsequent automated processing using specialized software [24,56].

The following features of cases can be defined:

- They represent domain specific knowledge related to the context that allows one to use this knowledge at the application level;

- They can be presented in various forms, cover different time periods, and be associated with solutions with descriptions of problems, results, situations, etc.

Each case is a unit of experience, the structure of which depends on the specifics of a particular problem but includes two main parts:

- Identifying (characterizing), which describes the experience in a way that allows one to assess the possibility of its reuse in a specific situation;
- Training (solution), which is a solution to a problem, proof of a solution (conclusion), alternative, or unsuccessful solutions.

Solving problems with the use of CBR consists of a number of main stages: retrieve, reuse (including adaptation), revise, and retain, which are preceded by the stages of conceptualization and the formation of a case base.

The task of the first preparatory stage (the conceptualization stage) is to form a case model based on some domain model describing a problematic situation. In our case, a problematic situation can be either a forest fire or a forest quarter. In fact, the case model is formed by redistributing significant characteristics or concepts of the domain according to the parts discussed above: identifying and training. As a result, a certain pattern is formed in the form of a domain class or frame [55,57]. In this paper, information on the main factors influencing the occurrence and development of a wildfire (see Section 2.1) were used to form a case.

At the second preparatory stage, the case base is formed according to the obtained model with possible indexing of cases [22]. To increase the speed when working with a case base, a database management system is usually used; however, typed text files can also be used for prototyping [55,57].

Retrieval is the first main stage, the main purpose of which is to extract one or more cases similar to the current problem (situation) from the case base. Various metrics are used to calculate similarity [20,22,55,57]: Euclidean, Hemming, probabilistic, Rogers–Tanimoto, Manhattan, Chebyshev, Mahalonobis, Zhuravlev, Bray–Curtis, Chekanovsky, Jacquard, etc. In this paper, we apply the modified Zhuravlev metric [58], which we used earlier [22]:

$$s_i(c^*, c_i) = \sum w_j h_j(p^*_j, p_{ij})/N, \tag{1}$$

where $s_i$ is a similarity; $c^*$ is a description of a new object or a new case (current situation); $c_i$ is an i-case from a case base; $w_j$ is an information weight of a j-property of compared cases; $p$ is a property of compared cases; $N$ is the amount of properties of the compared cases; $i$ is an index of cases in the case base; $j$ is an index of a case properties; $h_j$ is a function for calculating similarity between property values. This function has three different forms depending on a type of a property value: quantitative, qualitative, and interval.

For quantitative values:

$$\begin{aligned} h_j(p^*_j, p_{ij}) = 1, \ |v^*_j - v_{ij}| < \xi, \\ h_j(p^*_j, p_{ij}) = 0, \ |v^*_j - v_{ij}| \geq \xi, \end{aligned} \tag{2}$$

where $v^*_j$ is a value of a j-property of a new object or a new case (current situation); $v_{ij}$ is a value of a j-property of an i-case from a case base; $\xi$ is an acceptable difference in values, which is adjusted on the basis of empirical experimental data.

For qualitative values:

$$\begin{aligned} h_j\left(p*_j, p_{ij}\right) = 1, \ v*_j \in \{v_{j1}, \ldots v_{jM}\} \\ h_j\left(p*_j, p_{ij}\right) = 0, \ v*_j \notin \{v_{j1}, \ldots v_{jM}\} \end{aligned} \tag{3}$$

where $\{v_{j1}, \ldots, v_{jM}\}$ is a list of qualitative values.

For interval values:

$$h_j\left(p*_j, p_{ij}\right) = 1, \; v*_j \in \left[v^+_j, v^{++}_j\right]$$
$$h_j\left(p*_j, p_{ij}\right) = 0, \; v*_j \notin \left[v^+_j, v^{++}_j\right]$$

(4)

where $v^+_j$ and $v^{++}_j$ are the boundaries of the intervals of the values of the case properties.

Reuse consists of using information from retrieved cases to solve a new problem. If necessary, the extracted information is adapted. In this paper, adaptation is not required; the main purpose of reuse is to obtain a number of cases with a certain similarity.

Revise implies checking the obtained solution by an expert. Retain is aimed at preserving the obtained solution in a case base to expand it. This is how training of the CBR systems takes place. When solving the problem of forecasting wildfires, the CBR revise and retain stages were not implemented.

As noted above, CBR provides effective use of accumulated experience in the absence of in-depth knowledge, allowing one to solve problems based on superficial knowledge, and can be considered as an intermediate method when moving to deeper research. In addition, the advantage of the method is the fact that the case base can be continuously updated as new wildfire data become available, while there is no need to train the model and select model parameters; there are also no problems of retraining it, etc.

## 3. Forecasting the Risk of Forest Wildfires Using a Case-Based Approach

Let us consider the task of forecasting the risk of forest wildfires using a case-based approach, including formalization, conceptualization, and the creation of a prototype of a case base.

### 3.1. The Study Area

The study area is Irkutsk Oblast—the largest oblast of the Russian Federation. Its area is 774,846 km$^2$. (4.52% of the territory of Russia), with a population of 2,357,134 people (2022) and population density of 3.04 people/km$^2$ (2022). Irkutsk Oblast is located in Eastern Siberia. The extreme southern point of the region is located at 51° north latitude, and the northern tip almost reaches the 65th parallel. From north to south, the region stretches for almost 1450 km and from west to east for 1318 km. The southeastern border of Irkutsk Oblast runs along Lake Baikal. The region occupies the southeastern part of the Central Siberian Plateau, the plateaus and ridges of which have heights from 500 to 1000 m. Forests occupy 78% of the territory [13]. There are 32 administrative districts in Irkutsk Oblast. In recent years, there has been an increase in the area of wildfires in the region (Figure 1). Mass wildfires have caused a significant change in the landscapes of the catchment area of Lake Baikal and a decrease in forest reserves. The largest areas of wildfires for an average of 10 years are distinguished in Katanga, Kiren, Mamsko-Chui, Bodaibinsk, Bokhansk, Ust-Kut, and Chunsk districts [59].

Further in the paper, the research results are considered for the territory of Bodaibinsk and Kazachinsk-Lena districts. Bodaibinsk district, with an area of 92 thousand km$^2$, is located on the Vitimo-Patomsky Highlands, in the northeastern part of Irkutsk Oblast, in an area equated to the regions of the Russia's Extreme North. The Visimskiy Nature Reserve with an area of 11.61 thousand km$^2$ is located in this area. The Kazachinsk-Lena district occupies most of the Pre-Baikal depression and the northern part of the Baikal ridge (1200–2000 m above sea level). The natural zone is the taiga, an area of 33 thousand km$^2$.

### 3.2. Data

The database includes more than 45,000 records describing information about thermal points identified as a result of analysis of satellite images for 2017–2020. This information is used as initial data. A thermal point is a significant increase in temperature on the earth's surface registered at the time of the satellite's passage in comparison with neighboring sites. A thermal point can indicate burning of garbage, a man-made process, a fall, a man-made

fire, or wildfire. A thermal point is fixed by the satellite in the form of polygon-type objects (a set of quadrilaterals) (Figure 2).

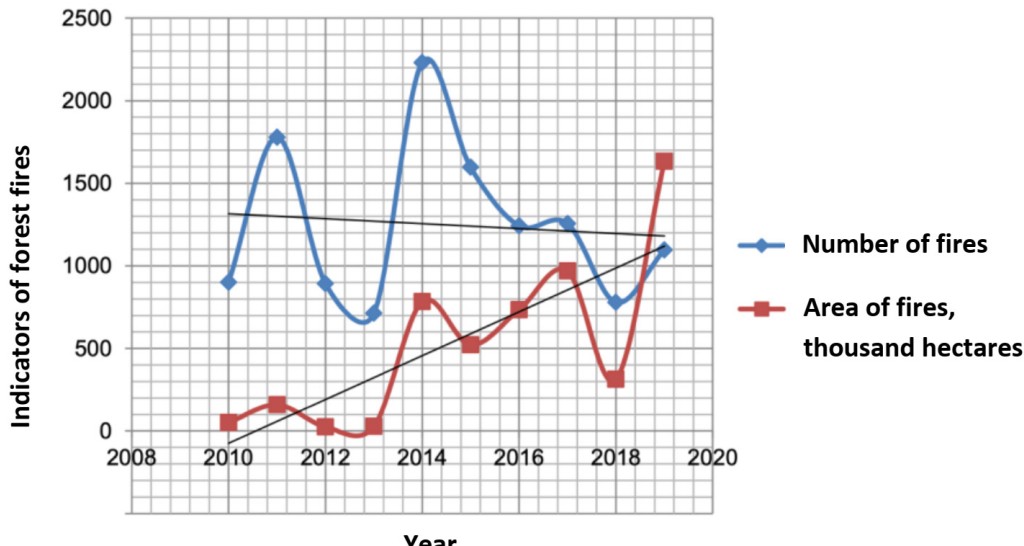

**Figure 1.** Statistics of forest fires in Irkutsk Oblast [59].

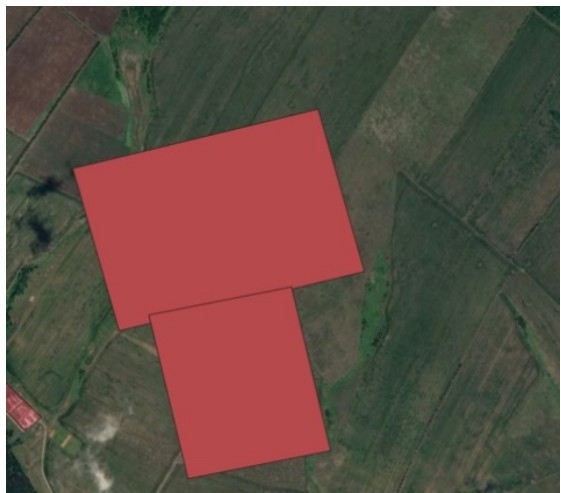

**Figure 2.** Example of a thermal point.

Information about thermal points is generated using software and hardware, which includes a satellite telemetry reception unit. The unit consists of a data reception station from NOAA Alisa-SC series satellites manufactured by Scanex LLC and specialized software for receiving satellite telemetry. Thus far, the satellite complex allows one to receive and decrypt data from the AVHRR device of the NOAA18 and NOAA19 satellites. The AVHRR characteristics are given in Table 1.

**Table 1.** The AVHRR characteristics.

| Satellite | Range, Microns | Spatial Resolution, m | Field of View, km | Repeatability of Shooting Single Territory |
|---|---|---|---|---|
| NOAA18, NOAA19 | 0.58–0.68; 0.725–1.0; 3.55–3.93; 10.3–11.3; 11.4–12.4 | 1100 | ~3000 | 3–4 times a day |

The visibility zone of the Alice-SK satellite complex extends from the Urals in the west to the Far East in the east of the Russian Federation, which provides monitoring of the fire-hazardous situation throughout Irkutsk Oblast.

The first stage of the monitoring system is direct reception of data from the NOAA satellite via the Alice-SK satellite complex, and then, several stages of data processing are carried out (demodulation, synchronization, decoding, etc.). At the final stage, analysis is performed by the FirePro software, ver.1 ISTP SB RAS, Irkutsk, Russia (Institute of Solar-Terrestrial Physics of the Siberian Branch of the Russian Academy of Sciences) [60–62] to identify thermal points (Figure 3), where the data obtained in automatic mode are further corrected by the operator to reduce possible errors in the algorithm and stored in the database that provided information for this study. At the same time, information about all linear and areal objects (thermal points, roads, rivers, etc.) is stored in the WKB (well-known binary) format.

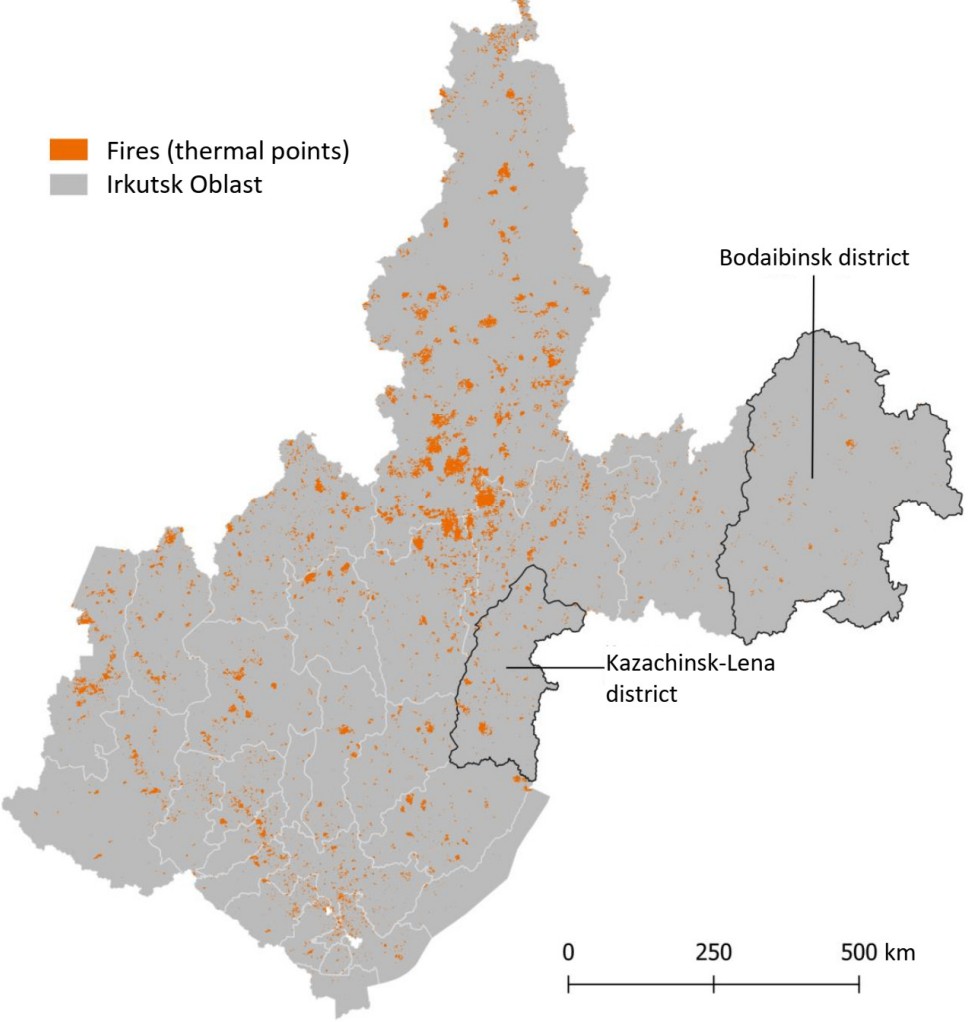

**Figure 3.** A map of Irkutsk Oblast displaying thermal points for 2017–2020.

### 3.3. Results (Employed Works)

Our research included stages and steps that can be presented in the form of the following sequence (Figure 4):

- Data preprocessing. This stage's primary goal is the transition from thermal points to forest fires;
- Formalization. This stage's main result is the structure of a case model;

- Conceptualization. This stage is dedicated to specifying a case model for the domain. In our case, the domain is the forecasting the risk of forest wildfires for Irkutsk Oblast;
- Prototyping a case base. This stage's primary goal is to create and debug a case base based on information about fires and a specified case model;
- Forest fire risk forecasting. This stage's main result is fire risk assessments for forest quarters.

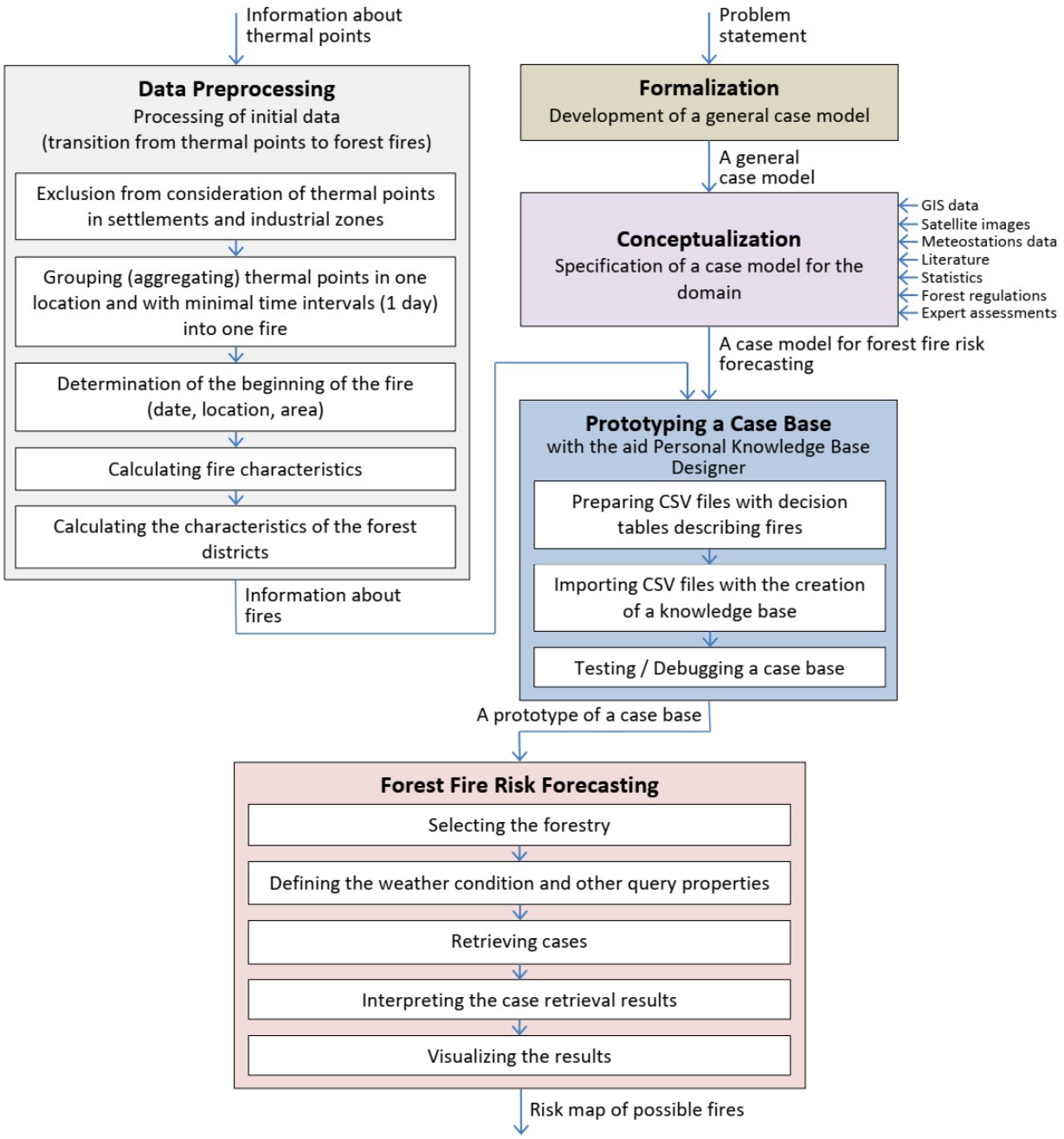

**Figure 4.** The main stages and steps of our research.

The first stages can be performed in parallel. Next, the stages are presented in more detail.

### 3.3.1. Data Preprocessing

The initial data in the form of 45,000 records describing information about thermal points includes records of daily (or with a break of up to three days) statistics about the

date and time of detection of the thermal point, its area, coordinates, and contours in the form of a polygon.

This initial data were preprocessed, including:

- Thermal points located within the boundaries of industrial zones, populated areas, and mining zones that are not forest fires were identified, with their subsequent exclusion from consideration. The remaining thermal points were later interpreted as forest fires.
- Grouped (aggregated) information about fires is gathered by combining data characterized by the location of the fire on intersecting polygons and the minimum time interval, with the determination of the duration, minimum, and maximum area of the detected fire.
- The initial events (ignitions) of a forest fire are highlighted with the determination of the date of detection, coordinates, and the fire polygon.
- Calculation of characteristics of each fire:
  a.　Area of each fire;
  b.　Municipal formation, forestry, forest area, forest quarter on whose territory a fire was detected;
  c.　Fire hazard class of the forest of the allocated territory;
  d.　Fire hazard class of the forest of the allocated territory according to weather conditions;
  e.　Forest-growing zone and forest-seed zoning area;
  f.　Weather conditions based on the data of the meteorological stations closest to the fire;
  g.　Data on the population density of municipalities affected by fires;
  h.　Distances from each fire to roads and railways as well as rivers and lakes and the nearest settlements; determination of the intervals of these indicators.
- Calculation of statistical indicators:
  a.　Number of fires and their areas in municipalities;
  b.　Number of fires and their areas in forest zones;
  c.　Number of fires and their areas in forest districts.

To implement the processing, the GeoAnalytics software module [63] was developed in Python3 using the "shapely", "pyproj", and "pandas" libraries to work with geo objects in the WKT (well-known text) and WKB (well-known binary) formats.

WKT is a text format for representing vector geometry and describing coordinate systems. To store the same information in the database, a binary equivalent format is used—WKB. The formats are part of the "Simple Feature Access" technology created to provide a common interface for the exchange of spatial data between various programs and services.

In particular, the "shapely" library used the WKB and WKT methods to load data. Further, to obtain the area of the polygon in square kilometers, the polygon was converted to a geodetic projection using the "pyproj" library, and the "Geod" class with the "WGS84" parameter and the "geometry_area_perimeter" method was used. In turn, the "pyproj" library and the "CRS" and "Transformer" classes as well as the "shapely" library and the "transform" method were used to obtain the projection of polygons (geometric data/geometric figure in radians) into geodesic quantities (geometric figure in meters). To obtain the distances, the "distance" method was used.

### 3.3.2. Formalization

Most decision-making tasks, including the current one, can be described by a set of characteristics and formalized as follows:

$$M^{Task} = (C, P, T, V), \tag{5}$$

where $M^{Task}$ is a model of the task; $C = \{c_1, \ldots, c_K\}$ are objects of some domain; $P = \{p_1, \ldots, p_N\}$ are properties of these objects; $T = \{t_1, \ldots, t_L\}$ are types of values, in particular, quantitative, qualitative, and interval; V is a set of property values of a certain type, i.e., they are given either as belonging to a certain type $p_j = \{v_{jm} \in T\}$, where $v_{jm}$ is m-value of j-property, T is the interval $[v^+_j, v^{++}_j)$, or a list of verbal values $\{v_{j1}, \ldots, v_{jM}\}$, or as a pair $p_j = \{v_{jm} \in T, \xi_j\}$; T is a quantitative/numeric type, and $\xi_j$ is a restriction on the permissible difference of values taken into account when comparing.

The task model from the point of view of the case-based approach is defined as follows:

$$M^{Task\_CBR} = Problem^{CBR} \rightarrow Decision^{CBR}, \tag{6}$$

where $Problem^{CBR}$ is a description of the current task (identifying part); $Decision^{CBR}$ is a description of the solution to the problem (training part), while $Problem^{CBR} = (c^*, C)$, $c^* \notin C$, where $c^*$ is a description of a new object or a new case (current situation); C is a description of a set of objects or a case base.

$$Decision^{CBR} = \{d_1, \ldots, d_r\}, \; dr = (c_r, s_r), \; c_r \in C, \; s_r \in [0; 1] \tag{7}$$

where $d_r$ is a description of the solution to the problem in the form of precedents extracted from the case base with an assessment of their similarity $s_r$ to the current situation $c^*$.

Next, this model was filled with domain information.

### 3.3.3. Conceptualization and a Case Model

To conceptualize and form a case model, we used the above-mentioned works in this area as well as pre-processed data from the database on fires in the Baikal natural territory for 2017–2020, information on weather, infrastructure (roads, settlements, etc.), and vegetation type.

In particular, the domain concepts and the relationships between them were defined (Figure 5). Based on the developed conceptual model, a case structure was created, which includes the following properties in the descriptive part: date and time of the start (detection) of a fire; fire territory; fire area; forest type; fire hazard class of a forest area; fire hazard class in the forest according to weather conditions; distance to motorways and railways; distance to the reservoir; distance to the nearest settlement; weather conditions: air temperature (T), atmospheric pressure (Po), wind direction (DD), wind strength (Ff), description of precipitation (type and level of precipitation), etc. The following properties were included in the training part of the case: forest quarter and assessment of the possibility of fire. A detailed description of properties is provided in Table 2.

**Table 2.** Properties of a case model.

| Case Properties | Description/Possible Values/Units of Measurement | Source of Information |
|---|---|---|
| | *Problem description* | |
| Date and time of the start (detection) of the fire | | Satellite images |
| Fire territory | Polygon | Satellite images |
| Fire area | Square kilometers | Satellite images |
| Forest quarter | Name of the forestry, plot, quarter | GIS data |
| Forest-seed zoning zone | Pine, spruce, larch, cedar | Forest regulations |
| Forest growing area | Forest-steppe, taiga, South Siberian mountain | Forest regulations |
| Fire hazard class of the forest area | High—I, above average—II, average—III, below average—IV, low—V | Forest regulations |

**Table 2.** *Cont.*

| Case Properties | Description/Possible Values/Units of Measurement | Source of Information |
|---|---|---|
| Fire hazard class of the forest according to weather conditions | V—extreme (more than 10,000), IV—high (from 4001 to 10,000), III—medium (from 1001 to 4000), II—small (from 301 to 1000), I (up to 300) | Determined by the complex indicator of V. G. Nesterov [64] |
| Distance to the nearest railway | Kilometers, intervals of values: 0–0.5; 0.5–3; 3–10; 10–15; more than 15 | GIS data |
| Distance to the nearest motorway | Kilometers, intervals of values: 0–0.5; 0.5–3; 3–10; 10–15; more than 15 | GIS data |
| Distance to the nearest reservoir | Kilometers, intervals of values: 0–0.5; 0.5–3; 3–10; 10–15; more than 15 | GIS data |
| Distance to the nearest settlement | Kilometers, intervals of values: 0–0.5; 0.5–3; 3–10; 10–15; more than 15 | GIS data |
| Population density | Number of inhabitants per square kilometer | Statistics of Irkutsk Oblast |
| Air temperature (T) | Degrees Celsius | Meteostations data |
| Dew point temperature (Td) | Degrees Celsius | Meteostations data |
| Atmospheric pressure (Po) | Millimeters | Meteostations data |
| Relative humidity of the air (U) | % | Meteostations data |
| Wind direction (rumba) (DD) | West-southwest; north; west-southwest; Calm, etc., meters per second, quantitative values were converted into qualitative ones on the 12-point F. | Meteostations data |
| Wind strength (Ff) | Beaufort scale: calm (0–0.2), quiet (0.3–1.5), light (1.6–3.3), weak (3.4–5.4), moderate (5.5–7.9), fresh (8.0–10.7), strong (10.8–13.8), etc. | Meteostations data |
| The amount of precipitation (RRR) | Millimeters, quantitative values were converted into qualitative ones: light rain (0.0–2), rain (3–14), heavy rain (15–49), very heavy rain (more than 50), etc. | |
| Description of precipitation | Heavy rain(s) light rain(s) during the observation period or in the last hour, etc. | Meteostations data |
| Dry thunderstorm | Yes, no | Meteostations data |
| Snowiness of winter | Low snow, norm, multi-snow | Meteostations data |
| *Decision description* | | |
| Forest quarter | Name of the forestry, plot, quarter | GIS data |
| Fire risk assessment | The value determining the risk of losses from a forest fire | Calculated based on statistical data |
| Assessment of the possibility of fire | Improbable, least probable, unlikely, probable, most probable | it is determined using the analysis of cases: improbable (very low)—no cases, least probable (low)—cases with similarity from 0 to 0.5, unlikely (moderate)—cases with similarity from 0.5 to 0.8, probable (high)—from 1 to 5 cases with similarity more than 0.8, most probable (very high)—more than 5 cases with similarity more than 0.8. |

A digital map of the area at a scale of 1:200,000 is used as an electronic topographical basis. The main thematic layers are vector layers obtained from various institutions. Data on the water system of Irkutsk Oblast, roads, and railways were provided by the Institute of Geography SB RAS [65]; data on forestry and forest quarters were obtained from the materials of the state forest inventory; meteorological data were acquired from the Federal State Budgetary Institution "Irkutsk Department of Hydrometeorology and Environmental Monitoring" [66]; data on thermal points obtained from the Institute of Solar and Terrestrial Physics SB RAS [60].

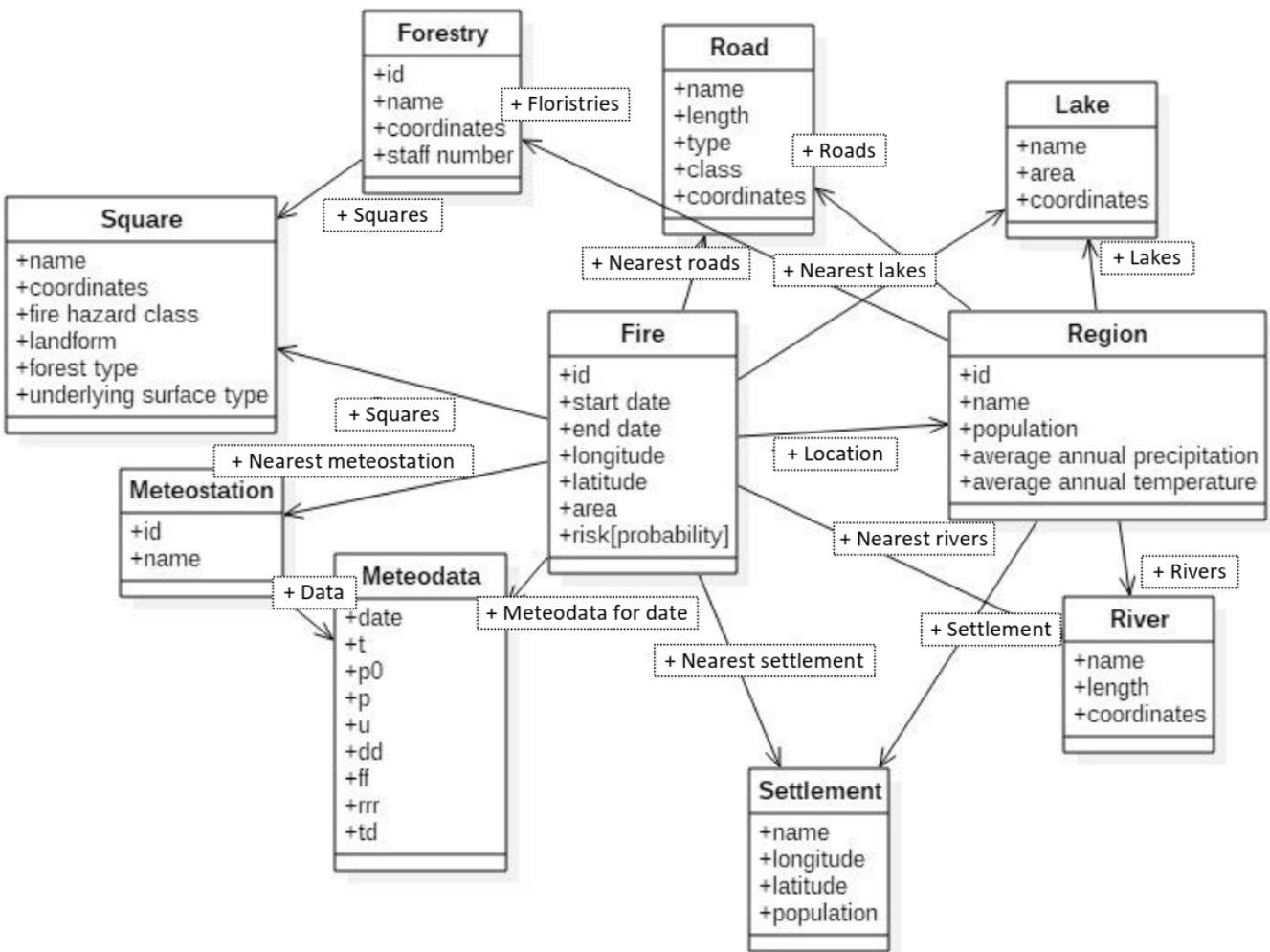

**Figure 5.** A fragment of the conceptual model of the domain.

In accordance with the model described, a data set was formed in the form of textual files of the CSV (comma-separated values) format intended for their further processing when creating a prototype of the case base.

The structure of the description of the current case c* corresponds to the described case model. However, as the object of the situation, we considered not the forest fire but the state of each forest quarter of a territory at a certain point in time. The choice of a forest quarter as a spatial unit for data processing is due to the limited data available. In particular, data on forest type and fire hazard class of a forest area are linked to forest quarters.

### 3.3.4. Prototyping a Case Base

The proposed approach was tested by creating and debugging a prototype of the case base for wildfire risk forecasting. The personal knowledge base designer (PKBD) [23] is our software that was used as a mean of prototyping.

Creating a prototype of a case base includes the following steps:

1. Preparing CSV files with decision tables for PKBD import;
2. Importing CSV files containing information for cases with the creation of a knowledge base;
3. Testing imported data by queries with subsequent correction of the data.

Below, we describe these steps in detail.

Step 1 involves preparing initial data in the form of CSV files for importing to PKBD. The form of decision tables supported by PKBD has the following features [67]: the ability to include a column with the names of rules to the table structure; the ability to indicate dependent columns with the symbol "#"; the ability to specify compound column names,

including the name of the entity (or class name) and the name of its property, separated by the string "::"; the absence of restrictions on the values in the cells; i.e., they can consist not only of a set of values {yes, no} representing a degenerate form of truth tables, but they allow one to use specific arbitrary values as cell values instead of only values indicating the presence or absence of a certain property (component) in the case structure.

In our case, the following table structure (headers) is used to determine the hazard class, which is formed based on the case model: id; fire id; new fire id; dt; lat; lon; fire territory; fire area; municipalities; population density; forestry; weather station id; weather station name; RRR (the amount of precipitation); Ff (wind strength); U (relative humidity of the air); T (air temperature); Td (dew point temperature); DD (wind direction); WW (current weather reported from the weather station); Po (atmospheric pressure); area; distance to the nearest motorway; distance to the nearest railway; distance to the nearest reservoir; forest zone; forest-seed zoning areas; weather hazard class; snowiness; snowiness uncertainty; name locality; name MO locality; municipalities locality; distance to locality; dry thunderstorm; #kv; #forest hazard classes. An example of a decision table is shown in Figure 6, with each row corresponding to a description of a separate case.

| | 1 | 2 | 3 | 4 | 5 | 6 | 7 | 9 | 10 | 11 | 12 | 13 | 14 | 15 | 16 | 18 |
|---|---|---|---|---|---|---|---|---|---|---|---|---|---|---|---|---|
| 1 | id | fire id | new fire id | dt | lat | lon | fire territory | municipali | populatio | forestry | weather s | weather s | RRR | Ff | U | Td |
| 2 | 74 | 18519 | 112 | 02.04.2020 10:47 | 56.5876 | 99.685 | 010600002C | Chunsk | 1.315849 | Chunsk | 29590 | Novoshun | none | calm | 36.0 | -4.8 |
| 3 | 76 | 18618 | 115 | 03.04.2020 10:34 | 52.0071 | 104.404 | 010600002C | Irkutsk | 79.984232 | Irkutsk | 30710 | Irkutsk | none | weak | 31.0 | -5.5 |
| 4 | 79 | 18621 | 118 | 03.04.2020 10:34 | 52.8266 | 104.898 | 010600002C | Ehirit-Bula | 5.816963 | Ust-Ordi | 30713 | Ust-Ordin: | none | light | 25.0 | -7.8 |
| 5 | 80 | 18612 | 119 | 03.04.2020 10:35 | 54.4423 | 99.4606 | 010600002C | Nizhneudi | 1.311747 | Tulun | 30507 | Ikey | none | calm | 83.0 | -7.4 |
| 6 | 83 | 18669 | 122 | 04.04.2020 3:56 | 52.9415 | 103.337 | 010600002C | Cheremhc | 9.295758 | Cheremh | 30617 | Cheremhc | none | light | 62.0 | -7.4 |
| 7 | 84 | 18687 | 123 | 04.04.2020 10:22 | 52.4611 | 104.017 | 010600002C | Angarsk | 187.10891 | Irkutsk | 30715 | Angarsk | none | light | 32.0 | -6.8 |
| 8 | 88 | 18690 | 126 | 04.04.2020 10:22 | 52.8021 | 104.878 | 010600002C | Ehirit-Bula | 5.816963 | Ust-Orda | 30713 | Ust-Ordin: | none | quiet | 40.0 | -5.2 |
| 9 | 91 | 18717 | 129 | 05.04.2020 10:11 | 52.5717 | 104.028 | 010600002C | Irkutsk | 79.984232 | Irkutsk | 30715 | Angarsk | none | light | 42.0 | -6.3 |
| 10 | 96 | 18707 | 134 | 05.04.2020 10:11 | 54.3827 | 101.908 | 010600002C | Kuytun | 2.643665 | Kuytun | 30603 | Zima | none | light | 34.0 | -10.1 |
| 11 | 97 | 18708 | 135 | 05.04.2020 10:11 | 54.1311 | 99.8199 | 010600002C | Tulun | 4.92524 | Tulun | 30507 | Ikey | none | calm | 72.0 | -4.2 |
| 12 | 98 | 18709 | 136 | 05.04.2020 10:11 | 54.1922 | 99.2666 | 010600002C | Nizhneudi | 1.311747 | Tulun | 29892 | Hadama | traces | light | 62.0 | -7.4 |

**Figure 6.** A fragment of the decision table with forest fire data (RRR, the amount of precipitation; Ff, wind strength; U, relative humidity of the air; Td, dew point temperature).

Step 2 allows one to create the structure of a knowledge base during import or just to add data to an existing one. In the first case, the titles of the table columns are used as the structure of cases. Figure 7 shows an example of the PKBD GUI after importing a list of cases and a structure (a template) of a case. In this step, the user can correct the existing cases or add new ones. During the import process, scales (ranges) of possible values for certain properties of cases are also formed.

Step 3 involves making queries to the case base obtained. When making queries, the importance of separate properties can be defined. The importance of properties or information weight is a subjective value that affects the assessment of similarity. The scale [1, 100] is used to determine it. As can be seen in Figure 7, case properties describing weather data, features of surface and vegetation, and proximity of infrastructure are involved in the retrieval (queries). The remaining parameters (lat–long, fire territory and district names, etc.) are used only as additional information when interpreting the results.

As a result of retrieval, the cases are ranked by the value of similarity. Figure 8 shows the results of a query in the form of a list of similar cases with the ability to drill down into its detailed comparative preview (Figure 9).

The resulting prototype of the case base contains information on 2240 wildfires in Irkutsk Oblast for 2017 and 2020. Next, we used the obtained case base and case-retrieval modules of PKBD for wildfire risk forecasting.

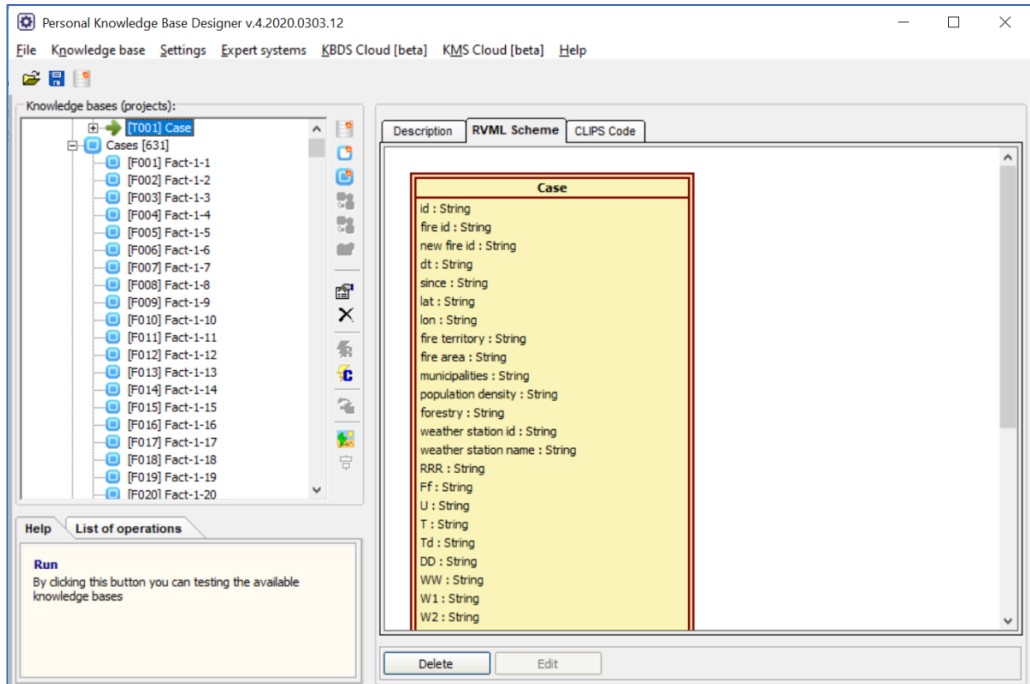

**Figure 7.** An example of a PKBD GUI form: description of a template (structure) for cases (631 forest fires for 2020). (dt, data; RRR, the amount of precipitation; Ff, wind strength; U, relative humidity of the air; T, air temperature; Td, dew point temperature; DD, wind direction (rumba); WW, current weather reported from the weather station; W1, past weather between observation dates 1; W2, past weather between observation dates 2).

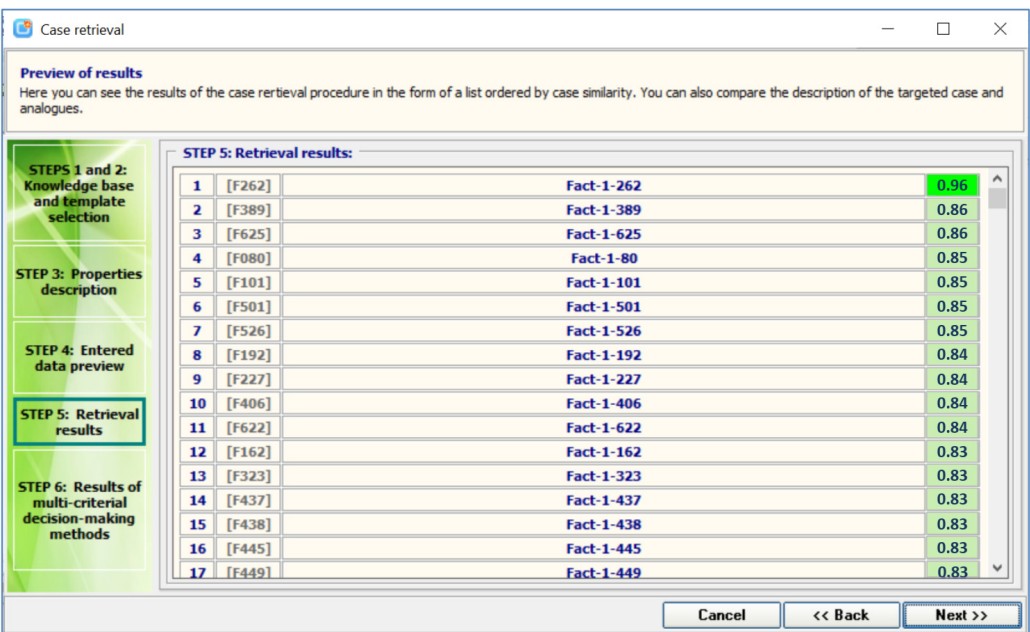

**Figure 8.** An example of a PKBD GUI form: case retrieval results.

**Figure 9.** An example of a PKBD GUI form: detailed preview of the CBR results. (RRR, the amount of precipitation; Ff, wind strength; U, relative humidity of the air; T, air temperature; Td, dew point temperature; DD, wind direction (rumba); WW, current weather reported from the weather station; W1, past weather between observation dates 1; W2, past weather between observation dates 2; Po, atmospheric pressure). To highlight the importance of values, a color indication is used: when comparing properties, close values are displayed in green, non-close values are displayed in red; when showing similarity, a high similarity is displayed on a green background, a pale pink background is used for a low one, and yellow and blue colors are also possible for intermediate values of similarity.

3.3.5. Forest Fire Risk Forecasting Based on CBR

To test the approach proposed, we selected two of the forest districts of Irkutsk Oblast, namely Bodaibinsk and Kazachinsk-Lena. First one includes 2818 separate quarters and the second 3695.

The forecasting risk (a hazard class) of forest fires for each quarter based on CBR contains the following stages:

(1) Defining the weather condition and other properties of certain forestry for a certain date: In particular, the following properties are used as the main ones: forest_hazard_classes, RRR (the amount of precipitation), Ff (wind strength), U (relative humidity of the air), T (air temperature), DD (wind direction), WW (current weather reported from the weather station), W1 (past weather between observation dates 1), W2 (past weather between observation dates 2), Po (atmospheric pressure), weather_hazard_class, snowiness, forest_zone, forest_seed_zoning zones, and thunderstorm; then, we used these data as a problem description (query) for case retrieval.

(2) Retrieving cases using our prototype of case base and software: The case retrieval procedure uses the similarity metrics presented in Section 2.3 and is divided into two sub-steps: (2.a) calculating the similarity for each case and (2.b) summing the retrieved cases by similarity intervals, which are then used to interpret the results.

As a result of (2.a), a detailed description of the calculated similarities is formed, indicating a certain variable (property) and its contribution to the total similarity (Figure 10). In this case, the final score is calculated as a weighted linear convolution (Equation (1)).

The result of (2.b) is a list of forest quarters characterized by information of a number of cases with different similarities. We used the following intervals for summing cases with accordance of a similarity: 0–0.5, 0.5–0.8, and 0.8–1 (Figure 11).

| | case id | forest_ha; | RRR | Ff | U | T | DD | WW | W1 | W2 | Po | weather_h | snowiness | forest_zor | forest_sec | thunderst | similarity |
|---|---|---|---|---|---|---|---|---|---|---|---|---|---|---|---|---|---|
| 1 | | | | | | | | | | | | | | | | | |
| 2 | 335 | 0.5 | 1 | 0.33 | 0.8244 | 0.947 | 0 | 1 | 1 | 1 | 0.988 | 1 | 1 | 0 | 0 | 0 | 0.799174495 |
| 3 | 337 | 0.5 | 1 | 0.33 | 0.8244 | 0.947 | 0 | 1 | 1 | 1 | 0.988 | 1 | 1 | 0 | 0 | 0 | 0.799174495 |
| 4 | 331 | 0.5 | 1 | 0.33 | 0.7643 | 0.810 | 0 | 1 | 1 | 1 | 0.998 | 1 | 1 | 0 | 0 | 0 | 0.783644614 |
| 5 | 354 | 0.5 | 1 | 0.33 | 0.7141 | 0 | 0.2 | 1 | 1 | 1 | 0.999 | 1 | 1 | 0 | 0 | 0 | 0.728656075 |
| 6 | 355 | 0.5 | 1 | 0.33 | 0.7141 | 0 | 0.2 | 1 | 1 | 1 | 0.999 | 1 | 1 | 0 | 0 | 0 | 0.728656075 |
| 7 | 356 | 0.5 | 1 | 0.33 | 0.7141 | 0 | 0.2 | 1 | 1 | 1 | 0.999 | 1 | 1 | 0 | 0 | 0 | 0.728656075 |
| 8 | 523 | 0.5 | 1 | 0.33 | 0.9783 | 0.642 | 0.2 | 1 | 1 | 1 | 0.997 | 0 | 1 | 0 | 0 | 0 | 0.72073095 |
| 9 | 418 | 0.5 | 1 | 0.33 | 0.9171 | 0.584 | 0.2 | 1 | 1 | 1 | 0.999 | 0 | 1 | 0 | 0 | 0 | 0.710952695 |
| 10 | 332 | 0.5 | 1 | 0.33 | 0.7997 | 0.687 | 0.2 | 1 | 1 | 1 | 0.973 | 1 | 0 | 0 | 0 | 0 | 0.707526302 |
| 11 | 424 | 0.5 | 1 | 0.33 | 0.9407 | 0.511 | 0.2 | 0 | 1 | 1 | 0.987 | 1 | 1 | 0 | 0 | 0 | 0.705796343 |

**Figure 10.** A fragment of the CSV file with the results of calculating similarity for a separate quarter. Each row corresponds to a specific case and similarities of its properties; the last column presents a total (or global) similarity.

| | kv_id | 0-0.5 | 0.5-0.8 | 0.8-1 | hazard class |
|---|---|---|---|---|---|
| 1 | | | | | |
| 2 | 3786 | 556 | 69 | 6 | I |
| 3 | 3787 | 532 | 90 | 9 | I |
| 4 | 3788 | 556 | 69 | 6 | I |
| 5 | 3789 | 501 | 120 | 10 | I |
| 6 | 3790 | 556 | 69 | 6 | I |
| 7 | 3791 | 550 | 81 | 0 | II |
| 8 | 3792 | 532 | 90 | 9 | I |
| 9 | 3793 | 539 | 92 | 0 | II |
| 10 | 3794 | 556 | 69 | 6 | I |
| 11 | 3795 | 532 | 90 | 9 | I |
| 12 | 3796 | 556 | 69 | 6 | I |

**Figure 11.** A fragment of the CSV file with the results of summing the cases by intervals of similarity.

The intervals are defined based on the opinion of experts. Since data on real fires are compared, it is assumed that if the similarity of cases tends to 1, then this fact significantly increases the probability of a wildfire.

The score value of 0.8 indicates that 20% of the signs of cases are allowed to mismatch when the current situation is assessed as dangerous. To determine statistical reliability, it is proposed to assess a high risk if there are more than five cases describing the current situation in which the similarity is in the range from 0.8 to 1.

An estimate value of 0.5 indicates that only 50% of the signs of cases coincide with the current situation. In this case, we assume that the risk of wildfire under these conditions is moderate.

Lower risk indicators are not identified at this stage of research.

In the future, when receiving new data and conducting experiments, the values of the intervals will be refined.

(3) Interpreting the case retrieval results using the following heuristic decision-making rules and defining hazard class according to the scale, where I is very high, II is high, III is moderate, IV is low, and V is very low:

a.　IF no cases, THEN forest fire risk is improbable (hazard class V);

b.　IF there are cases with a similarity from 0 to 0.5, THEN forest fire risk is least probable (hazard class IV);

c.　IF there are cases with a similarity from 0.5 to 0.8, THEN forest fire risk is unlikely (hazard class III);

d.　IF there are more than 50 cases with a similarity from 0.5 to 0.8, THEN forest fire risk is probable (hazard class II);

e.     IF there are at list one to five cases with a similarity more than 0.8, THEN forest fire risk is probable (hazard class II);

f.     IF there are more than five cases with a similarity more than 0.8, THEN forest fire risk is most probable (hazard class I).

In Figure 11, the interpretation results are presented in the last column.

(4) Visualizing the results of the interpretation on the maps (Figures 12 and 13):

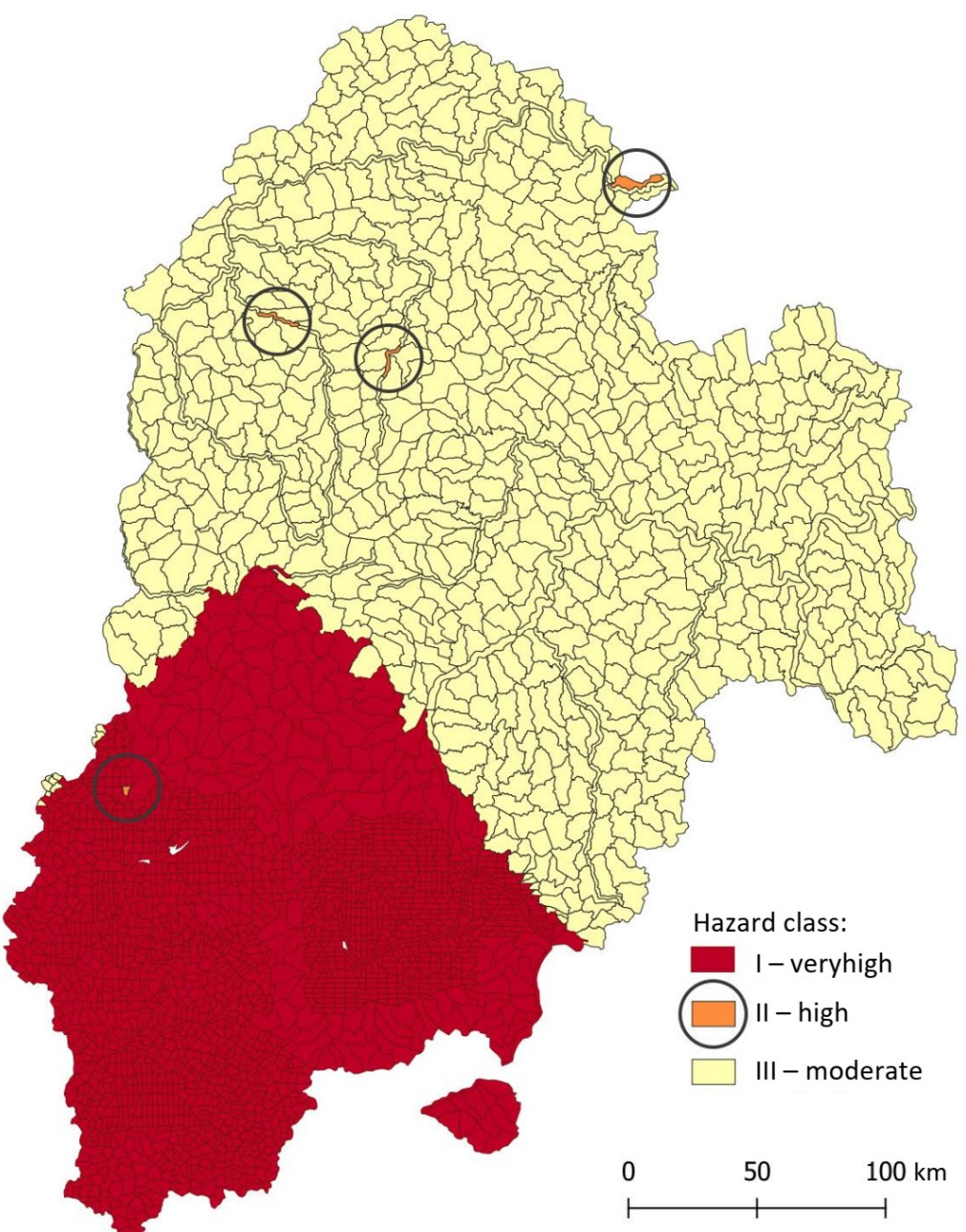

**Figure 12.** The forecasting results in the Bodaibinsk forestry.

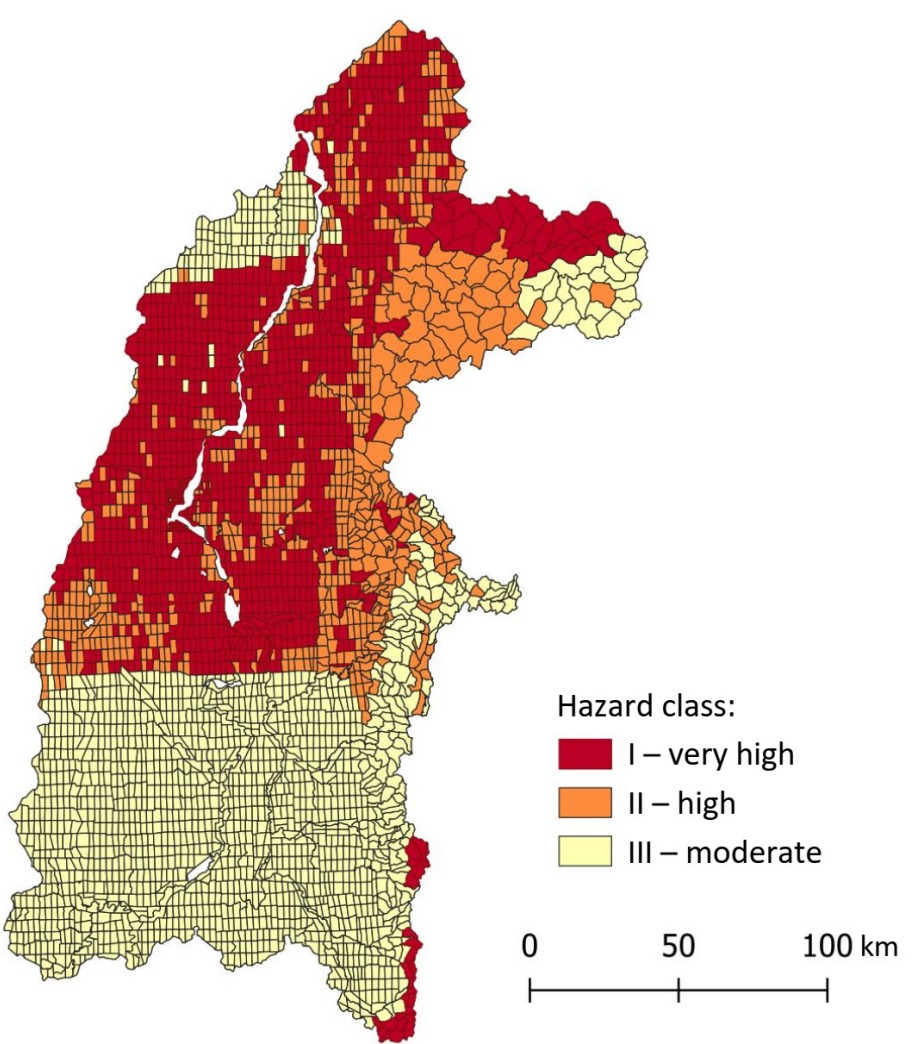

**Figure 13.** The forecasting results in the Kazachinsk-Lena forestry.

## 4. Discussion and Conclusions

The analysis of the CBR results showed a different contribution of the variables (properties) to the formation of similarity of the retrieved cases (Table 3). For this purpose, the quarters that received the highest scores and ensured the formation of I and II hazard classes were analyzed.

According to these data (Table 3), it follows that the factors describing weather conditions made the greatest contribution to the formation of the total similarity. Their cumulative contribution was 63.8%. Next are the factors characterizing the state of the forest. Their total contribution was 36.2%.

Thus, it can be argued that one of the reasons for the predominance of high-risk quarters in the southern part of the Bodaibinsk forestry and in the central and northern parts of the Kazachinsk-Lena forestry when vegetation differs is a small number of weather stations. This fact led to the use of the same weather data for a certain group of quarters.

We evaluated the obtained CBR results by their comparison with the results of real monitoring for a certain time period. For our evaluation, we used the following accuracy score based on the calculation of a delta (difference) between CBR and monitoring values for forest quarters:

$$\begin{aligned} &\text{Accuracy} = 1, \, n^{CBR} \geq n^{F}; \\ &\text{Accuracy} = n^{CBR}/n^{F}, \, n^{CBR} < n^{F} \end{aligned}$$
(8)

where $n^F$ is a number of forest fires for the quarter with accordance of monitoring data; $n^{CBR}$ is a number of retrieved cases with a similarity more than 0.8 for the quarter; Accuracy is the accuracy score calculated in different ways with accordance to a $n^{CBR}$ value.

We calculated the accuracy score for quarters of the Bodaibinsk and Kazachinsk-Lena forest districts. The average score value reached 0.51.

Table 4 shows a comparison of the results obtained with the closest works [44,45] and others [6,8,14,47].

**Table 3.** The main variables (properties) influencing the formation of the similarity.

| Variable (Property) | Average Similarity between Separate Properties (Local Similarity) | Contribution to the Formation of the Total (Global) Similarity, % |
|---|---|---|
| The amount of precipitation (RRR) | 1 | 9.65 |
| Fire hazard class of the forest according to Weather conditions | 1 | 9.65 |
| Snowiness of winter | 1 | 9.65 |
| Forest hazard classes | 1 | 9.65 |
| Forest zone | 1 | 9.65 |
| Atmospheric pressure (Po) | 0.99 | 9.55 |
| Air temperature (T) | 0.92 | 8.88 |
| Current weather reported from the weather station (WW) | 0.75 | 7.23 |
| Forest-seed zoning zones | 0.75 | 7.23 |
| Relative humidity of the air (U) | 0.63 | 6.08 |
| Past weather between observation dates 1 (W1) | 0.33 | 3.18 |
| Past weather between observation dates 2 (W2) | 0.33 | 3.18 |
| Wind strength (Ff) | 0.33 | 3.18 |
| Wind direction (rumba) (DD) | 0.33 | 3.18 |

In particular, in [44,45], authors used different datasets (case bases) and case models. In [44], they achieved 68.8% forecast effectiveness for a dataset of 560 cases. In [45], the use of CBR provided 58% of good predictions (which can be interpreted as an accuracy score) for a dataset with 2000 cases (which is commensurate with the case base in our work). At the same time, the combination of CBR with other technologies in [45] achieved good predictions of 66%. Other techniques with higher scores are used by researchers in [6,8,14,47]. A more detailed consideration of approaches and assessments is presented in [18]. In this connection, our results look modest, but our work can be considered as a first stage of a larger investigation within the project "Fundamentals, methods and technologies of digital monitoring and forecasting of the ecological situation of the Baikal natural territory". In the future, using the experience of other researchers [8,18,45], we plan to increase the effectiveness of forecasting through the development of the case model as well as through the joint use of various technologies such as case-based, rules-based reasoning and machine learning methods.

The solution considered in this paper has its limitations, disadvantages, and advantages. In particular, an important limitation is the availability of information on forest quarters and real fires. Therefore, in this work, the Bodaibinsk and Kazachinsk-Lena districts were selected since it was possible to form a complete dataset for them.

The study also revealed a weakness in the coverage of Irkutsk Oblast by weather stations, which led to the use of the same weather data for groups of quarters, which generally lowered the accuracy of forecasting.

A correlation analysis of variables (properties) was also not carried out in order to reduce the set and exclude dependent variables from processing, while the expediency of using aggregated indices was determined, as in, for example, [44].

The advantages of the approach proposed can be considered from two points of view: (1) the creation and debugging of case bases and (2) solving the task of forecasting the risk of forest fires.

Table 4. A comparison with existing approaches/methods.

| Work/Criteria | Area Name | Area Features | The Main Fire Reason | The Volume of the Case-Base or Training Sample | Number of Case Properties or Factors | Technique | The Accuracy or Prediction Score |
|---|---|---|---|---|---|---|---|
| **Our research** | **Irkutsk Oblast (Russia)** | **Low population density; taiga is a overexposed vegetation type; a small number of weather stations** | **Weather and forest condition** | **2240** | **41 (38/2) [a]** | **CBR** | **0.51** |
| [44] | DaXingAnMountains (China) | Hard-to-reach places; a small number of weather stations | Careless handling of fire when visiting the forest; lightning (dry thunderstorms) | 560 | 5 [b] | CBR | 0.688 |
| [45] | Mediterranean areas | - | - | 2000 | 20 (10/10) [c] | Radial Basis Function (RBF) Network<br>CBR<br>CBR + RBF | 0.52<br>0.58<br>0.66 |
| [14] | Noshahr forests (Iran) | A popular tourist destination | Careless handling of fire when visiting the forest | 30 | 10 | Analytical NetworkProcess (ANP) and Fuzzy Logic (FL) | 0.819 |
| [6] | Huichang County (China) | Long fire season; the presence of meadows | - | 244 (122 + 122) [d] | 10 | Weights-of-Evidence (WOE) model and a knowledge-based AnalyticalHierarchy Process (AHP)<br>Logistic regression | 0.91<br><br>0.77 |
| [47] | Minudasht Forests (Iran) | Tropical dry forests is a overexposed vegetation type | - | 151 | 7 | Modified AHP<br>Mamdani FL | 0.777<br>0.882 |
| [8] | Pakistan | Arid climate | - | 44,622 (22,311 + 22,311) [e] | 34 | Machine learning (8 algorithms) | 0.577–0.852 |

[a] 38 properties in the identifying part and 2 properties in the training part; [b] three dynamic and two static calculated indexes were used to describe the cases; [c] 10 properties in the identifying part and 10 properties in the training part (the same properties but with the values of the prediction step); [d] 122 recent fires + 122 random no-fire points from the fire-free areas; [e] 22,311 fire points + 22,311 no-fire points.

From the first point of view, the proposed use of decision tables and PKBD contributes to the involvement of end users in the development process and ensures the creation of prototypes of knowledge bases and their debugging without involving programmers. According to the estimates of the effectiveness of this approach for rule-based knowledge bases from [68], the use of decision tables can reduce the time spent compared to hand-coding by an average of 52%.

From the second point of view, the use of CBR allows one to use domain models (Figure 5), providing their mapping into the structures of the knowledge base, and also makes the decision-making process "transparent" since the mechanisms of forming the similarity are known, and when it is detailed, it is possible to determine variables of greater importance for a certain situation.

Precise and timely forecasting of forest fire risks provides effective decision support and situational control. In this work, we proposed the use of case-based reasoning for solving this task. Our results connected the formalization and conceptualization of the domain, building a case model, prototyping a case base, and evaluation. At the same time, the case model reflects the basic concepts and relationships describing the main characteristics of forest fires; in turn, the case base accumulated information about 45,000 thermal points in Irkutsk Oblast for 2017–2020.

The efficiency of the approach proposed is evaluated from a formal and substantive point of view using the example of the Bodaibinsk and Kazachinsk-Lena districts. Based on the results of the assessment, it can be concluded that it is necessary to use a set of different methods (data mining, neural networks) for more accurate forecasting. In this context, the use of cases helps create training samples and ensures the formation of preliminary decisions, which can be further refined.

The application and elaboration of other methods to solve the forecasting problem as well as the refinement of the web service are the directions of further research.

**Author Contributions:** Conceptualization and methodology, O.N. and A.Y.; software, A.Y. and N.D.; validation, A.Y., J.P. and O.N.; writing—original draft preparation, A.Y.; visualization, J.P. All authors have read and agreed to the published version of the manuscript.

**Funding:** This work was supported by the Ministry of Science and Higher Education of the Russian Federation, Grant No. 075-15-2020-787 for implementation of Major scientific projects on priority areas of scientific and technological development (the project "Fundamentals, methods and technologies for digital monitoring and forecasting of the environmental situation on the Baikal natural territory").

**Institutional Review Board Statement:** Not applicable.

**Informed Consent Statement:** Not applicable.

**Data Availability Statement:** Not applicable.

**Conflicts of Interest:** The authors declare no conflict of interest.

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
