# Peer review of "Forest Fire Risk Forecasting with the Aid of Case-Based Reasoning"

_applsci, doi:10.3390/app12178761_

Round 1
Reviewer 1 Report
The authors discuss the main stages of solving the problem of forecasting the risk of forest fires based on a case-based approach. The idea is interesting and the results are well discussed. However, some questions should be addressed.
Q1. Make sure all abbreviations are written out in full the first time used. Please check the manuscript carefully.
Q2. English should be polished.
Author Response
Dear reviewer! Thanks a lot for your comment. We have thoroughly analyzed all of them and significantly revised our paper. From a formal point of view our paper extended from 12 up to 19 pages, and our reference list was extended from 27 up to 66 positions, all main changes in the text are highlighted in red.

Reviewer 2 Report
The authors discuss the forecasting the risk of forest fires based on a case-based approach. The main stages include data preprocessing, case model and the prototype creation based on expert system. The results look encouraging and motivating. But there are still some contents, which need be revised in order to meet the requirements of publish. A number of concerns listed as follows:
(1) The abstract does not provide significant information and it should be revised to highlight the significant methodological contributions and conclusions.
(2) The theoretical background of the proposed method is adequately detailed in the paper.
(3) The inspiration of your work must further be highlighted in introduction. Some suggested recent literatures should add. For example, 10.3390/agriculture12060793; 10.1109/JSTARS.2021.3059451; 10.1016/j.engappai.2022.105139; 10.1007/s10489-022-03719-6 and so on.
(4) To explore Comparative results with existing approaches/methods relating to the proposed work.
(5) How to set up Case Base in Section 3.4. How to set the scale of cases?
(6) In Line 210, CSV, RRR, DD, WW… are what meanings?
(7) Figure 3 and Figure 4 are not clear. Please revise them.
(8) Analysis is insufficient. An extensive analysis is required.
(9) Result and discussion should be rewritten to summarize the significance of the work.
(10) Correct typological mistakes and mathematical errors.
Author Response

(The authors gave the same response as above.)

Reviewer 3 Report
Manuscript Title: Forest Fire Risk Forecasting with the aid of Case-Based Reasoning
Authors: Dorodnykh N., Nikolaychuk O., Pestova J., Yurin A.
General Remarks: Thank you for the invitation. I have carefully examined the manuscript, and I think that the manuscript requires some major revisions. The topic is very important and the manuscript contains some valuable output. It is within the scope of the journal and special issue. Title and abstract are concise. Therefore, this manuscript could be a valuable article. However, the current form of the manuscript lacks important content that should be presented in any sort of hazard risk forecasting study. I have listed my major concerns below. Furthermore, I found the introduction and background sections lacking a good design. I gave a detailed recommendation to the authors on how they needed to prepare their introduction and literature review. The methodology is novel for forest fire studies, but there are some issues related to the presentation of the methodology. Last but not least, I have some minor comments and a question at the end of my report. If the authors would like to have a publication with high impact and debate, my recommendations will help them improve the general quality of the paper. My recommendations may take some time, but they will definitely improve the content of your paper. Good luck.
Introduction and Background
1) First of all, according to my scan on iThenticate, there are many texts matched with a conference paper and an article published by the same authors. There are many, especially in Section 2. I guess this may lead to self-plagiarism, since the matched ratio exceeds the tolerated limit. Please try to paraphrase and summarize them.
2) Mentioning the project name at the very beginning of the introduction seemed a bit odd to me. You can introduce the motivation of your project a bit later (it may be located in your contributions list).
3) Line 58-62: I think the review on fire susceptibility / risk forecasting studies should be expanded. The authors claimed that the only disadvantage of other methods was to work with the individual characteristics of the studied territories. This fact may be true, but ‘locality’ cannot be expressed as a novelty or disadvantage. Susceptibility mapping / risk forecasting studies try to explain and define the forest fire ignition conditioning factors using different methods for different geographies. Your novelty is more methodological, not geographical. Please refer to the novel sources I listed below.
The literati try to determine the weights of the factors using statistical approaches (not static!) e.g., Multiple Criteria Decision Support Systems, weight-of-evidence, frequency ratio etc.
- https://doi.org/10.1080/19475705.2014.984247
- https://doi.org/10.1080/09640568.2019.1594726
- https://doi.org/10.1016/j.ecolind.2019.01.056
Some literati try to employ machine learning algorithms to create risk/susceptibility maps, and explain the contribution of conditioning factors by information gain, feature importance, or local and global XAI methods:
- https://doi.org/10.1016/j.ecoinf.2022.101647
- https://doi.org/10.3390/rs14081918
- https://doi.org/10.3390/inventions7010015
By elaborating on these methodologies, please express yourselves how your methodology (case-based reasoning) adds to, confirms, and contradicts the existing body of literature, mainly focusing on the determination of important conditioning factors. Why do we need to be inclined to use case-based reasoning? What are the cons and pros?
A simple sentence like “various approaches are used…” will not be sufficient for a multi-disciplinary WoS-indexed journal. I guess you get my point.
4) Between lines 65-73, you listed dozens of conditioning factors and called them “main factors.” Can you give relevant references for these factors? How did you select them? I think you listed all the factors used in the literature. This part should be reconsidered.
5) Lines 89-90. You mentioned ‘a special knowledge’. Can you give more details about this?
Methodology, Data and Results (Section 3)
6) I think that this section should be separated into two main sections. One is for methods and data, the other is for the results (or employed works).
7) Lines 147-151: The authors mention a database containing 45,000 active fire pixels (thermal point is a vague term). However, they did not mention the satellite platform from which these active fire pixels are collected. Since the abstract and intro emphasize the use of remotely sensed products, this data description should be expanded (including the study area, Bodaibinsk should be mentioned here, areal extent, geographical coordinates, forest fire history etc.). Furthermore, although this database consists of pixels, I didn’t get what are the contours in a form of polygons? (line 153). You have a point-based database, then you mention polygons and areal calculations (line 166). This is confusing. Did you prepare your CSV file with rows for each polygon?
8) You need to mention the institutions where you collect forestry and meteorological data sets, and the sources for physical and anthropogenic factors (e.g. motorways, settlements, reservoirs etc.). Their scale/spatial resolution can be added to Table 1. Then, before delving into a CSV file generation, you need to mention how you overlay these spatial data sets (in GIS), select a grid resolution, handle the coordinate system and resample them. You need to blend the data description with the GIS procedure in the text more.
9) I think that intervals for similarity assessment (Line 265) should be explained. How did you adopt these intervals? You need to cite references or explain more in detail. On the other hand, I’m not satisfied with the usage of Hazard Class I, II, and so on. It is well-acknowledged to use very high, high, moderate, low, very low to define the risk classes.
10) For this sort of studies, it is better to report areal distribution of risk classes in the study area (xx% very high, yy% high and so on), and the distribution of training data (active fire pixels or polygons) coinciding with these risk classes (%xx pixels in very high class, and so on). The first will help in land management, the second will help to understand the reliability of training process.
Discussions and Conclusions
11) You just compared your results with the paper belonging to Mata et al. Please add some comparisons with the work of Liu et al (2016). Furthermore, this section also requires insightful comments on the reasons why the scholars can use case-based reasoning (this time supporting with your findings). Lastly, add your limitations of your study here.
Minor Comments
- Line 128, pm (not sub-scripted in text)
- Line 156, “… that are not forest fires” seems incorrect. “that are out of forest areas” or “that are excluded areas” are better alternatives.
- Line 163: “The initial event” doesn’t sound good. Try “ignition”.
- Line 167: ‘forest area’ is used twice.
- Line 221: Instead of {yes,no}, {fire, non-fire} can be a better option, I guess.
- Line 228: RRR; Ff; U; T; Td; DD; WW; W1; W2; Po. These meteorological factors need to be explained. What are they abbreviated for? You can add the abbreviations in your Table 1.
- In Fig. 5, I see an orange color in the legend, but two similar red tones in the map. Control it please.
- I’m not quite sure whether you should give your accuracy metric in the discussion part (line 286). The literati generally present their performance metrics in methods.
Question (can be replied either in the manuscript or only in the response letter)
- In machine learning and statistical methods, the conditioning factors are firstly evaluated with a multi-collinearity test, and collinear factors are dropped before training/generating the models. In case-based reasoning, I see that you used similar data sets (such as lat-long, fire territory and district names semantically refer to the same location asset). Why do you need to use such similar data sets? How can you be sure that your processes are not overtrained? I see that your forecasting results shown in Fig.1 are fragmented homogenously. In such a large area, we need to see a more heterogeneous pattern since you use plenty of conditioning factors. The distribution of risk classes just like blocks seems a bit strange to me. Please elaborate this issue.
Author Response

(The authors gave the same response as above.)

Round 2
Reviewer 2 Report
According to the revised paper, I have appreciated the deep revision of the contents and the present form of this manuscript. There is little content, which need be revised according to the comment of reviewer in order to meet the requirements of publish. A number of concerns listed as follows:
(1) The authors need to interpret the meanings of the variables.
(2) Please highlight your contributions in introduction.
(3) To explore Comparative results with existing approaches/methods relating to the proposed work. The method/approach in the context of the proposed work should be written in detail.
(4) Conclusion: What are the advantages and disadvantages of this study compared to the existing studies in this area?
(5) The inspiration of your work must further be highlighted. Some suggested recent literatures in previous comments should add in the revised paper.
(6) Further correct typological mistakes and mathematical errors.
(7) More equations are necessary to explain the proposed method.
I hope that the authors can carefully and further revise this manuscript according to the reviewer comments in order to meet the requirements of publish.
Reviewer 3 Report
The revised manuscript shows an improvement, and the authors achieved a response to most of the queries (not all). The data description and GIS procedure are better explained, and the figures are OK.
Some issues addressed by me and other reviewer were not responded adequately. There are some wishful-thinking. For instance, risk class intervals are not still clear. Defining three intervals with arbitrarily selected thresholds seems pseudoscience. Scholars have tried a lot for finding a way to define these risk intervals using data-driven techniques or statistics (e.g., Jenks or quartiles). So, it’s up to authors’ decision and experiences. I think that a manuscript covering a risk forecasting should explain the understanding of risk more scientifically. Otherwise, our maps will be castles in the sky (semantically). Second, the authors avoid explaining the contribution of their conditioning factors to the fire risk, even though I recommended several readings to compare. An output like “The predominance of high-risk quarters in the southern part of the Bodaibinsk forestry and in the central and northern parts of the Kazachensk-Lena forestry is due to the location of the main settlements and road infrastructure in these places.” can be responded to by the literati with a query like “Aye, we all know that cigarettes and broken glasses thrown cause these fires, so why did you add the other factors in your analysis?” There should’ve been a more scientific approach to measure the impact of other factors. In its current form, the model is black-box.
To sum up, I recommend the acceptance of the current version. I don't need to see this again because I believe it is now of publishable quality in an acceptable level, but I do not believe the paper is scientifically very sound. I have no substantive feedback now in the sense of there being new work required or major outstanding queries still to be addressed. Congratulations.
